# Preparation of Biochar with Developed Mesoporous Structure from Poplar Leaf Activated by KHCO_3_ and Its Efficient Adsorption of Oxytetracycline Hydrochloride

**DOI:** 10.3390/molecules28073188

**Published:** 2023-04-03

**Authors:** Zhenhua Wei, Chao Hou, Zhishuo Gao, Luolin Wang, Chuansheng Yang, Yudong Li, Kun Liu, Yongbin Sun

**Affiliations:** Institute of Optical Functional Materials for Biomedical Imaging, School of Chemistry and Pharmaceutical Engineering, Shandong First Medical University & Shandong Academy of Medical Sciences, Taian 271016, China; zhwei@sdfmu.edu.cn (Z.W.); hczhentao1@163.com (C.H.);

**Keywords:** biochar, potassium bicarbonate, mesoporous structure, oxytetracycline hydrochloride, adsorption

## Abstract

The effective removal of oxytetracycline hydrochloride (OTC) from the water environment is of great importance. Adsorption as a simple, stable, and cost-effective technology is regarded as an important method for removing OTC. Herein, a low-cost biochar with a developed mesoporous structure was synthesized via pyrolysis of poplar leaf with potassium bicarbonate (KHCO_3_) as the activator. KHCO_3_ can endow biochar with abundant mesopores, but excessive KHCO_3_ cannot continuously promote the formation of mesoporous structures. In comparison with all of the prepared biochars, PKC-4 (biochar with a poplar leaf to KHCO_3_ mass ratio of 5:4) shows the highest adsorption performance for OTC as it has the largest surface area and richest mesoporous structure. The pseudo-second-order kinetic model and the Freundlich equilibrium model are more consistent with the experimental data, which implies that the adsorption process is multi-mechanism and multi-layered. In addition, the maximum adsorption capacities of biochar are slightly affected by pH changes, different metal ions, and different water matrices. Moreover, the biochar can be regenerated by pyrolysis, and its adsorption capacity only decreases by approximately 6% after four cycles. The adsorption of biochar for OTC is mainly controlled by pore filling, though electrostatic interactions, hydrogen bonding, and π-π interaction are also involved. This study realizes biomass waste recycling and highlights the potential of poplar leaf-based biochar for the adsorption of antibiotics.

## 1. Introduction

Oxytetracycline hydrochloride (OTC) is an effective broad-spectrum antibacterial agent, which plays an important role in animal husbandry and disease therapy [1]. However, approximately 70% of OTC is dispersed into the natural environment as feces and urine are excreted [2]. It has been demonstrated that the chronic exposure of zebrafish to OTC at a concentration of 0.42 g·L^−1^ will reduce its nonspecific immune response [3], and concentrations of 4 g·L^−1^ OTC can cause genotoxic damage to zebrafish [4]. The extensive use of OTC brings convenience to human life; however, its residues in the water environment lead to microbial death and antibiotic resistance gene transmission, which induce a threat to ecological and public health safety [5]. Moreover, OTC may cause endocrine disorders, cancer, and other diseases through the accumulation of food chain, which could seriously affect human health [6]. Therefore, it is of great importance to develop effective technologies to remove OTC from the water environment.

Presently, there are various technologies to remove antibiotics from the water environment [7], including biodegradation [8], chemical oxidation [9], membrane separation [10], photocatalytic degradation [11], adsorption [12], etc. Among these, the adsorption technology is simple, efficient, stable, and relatively economic [13]. More importantly, the adsorption process does not introduce secondary pollutants, and is considered to be one of the most effective methods for removing antibiotics in water [14,15]. Adsorbents are the key to adsorption technology [16,17,18,19,20]. In the past decades, researchers have developed a variety of adsorbents, including metal oxides [21], molecular sieves [22], MOFs [23], carbon nanotubes [24], MXenes [25], etc. Recently, biochar has been widely used to remove antibiotics from the environment due to its porous structure and low cost [26,27,28]. However, biochar obtained from the direct pyrolysis of biomass usually has a low specific surface area, undeveloped pore structure, and few adsorption sites, resulting in its poor adsorption performance [29,30]. In order to optimize the adsorption performance of biochar, activating agents are usually added during the synthesizing process. The most commonly used activators, such as KOH [31], H_3_PO_4_ [32], ZnCl_2_ [33], and FeCl_3_ [34], can effectively create pores in biochar, but they are strongly corrosive and cause serious pollution to the environment. Furthermore, the biochar synthesized with the above activators possesses a relatively large specific surface area, but lacks a mesoporous structure, which is not conducive to the improvement of adsorption rate or the adsorption of antibiotics with relatively large molecular sizes [35,36,37]. Therefore, it is more appropriate to choose an activator with less pollution and a greater mesoporous forming ability. Herein, we chose KHCO_3_ as the activator, as it can meet the two above-mentioned advantages. First, K_2_CO_3_ produced by KHCO_3_ during pyrolysis can react with the biomass to form microporous and mesoporous structures in the biochar. Second, CO_2_ gas produced by KHCO_3_ erodes the biomass and endows the biochar with more mesopores [38,39,40].

The biomass used to synthesize biochar are varied. Here, we choose poplar leaf as the raw material, which is widely distributed in China, Europe, West Asia, and North Africa, and has a huge annual output. The biochar is prepared from poplar leaf activated by KHCO_3_ using an oxygen-limited pyrolysis method. We studied the effect of the KHCO_3_ addition on the specific surface area and pore distribution of the biochar. In addition, the adsorption capability and adsorption mechanism of biochar to OTC was analyzed. Finally, we studied the adsorption performance of biochar on OTC in natural water matrices and its regeneration performance, which highlights the potential application of poplar leaf-based biochar in the remediation of antibiotic pollution.

## 2. Results and Discussion

### 2.1. Preparation and Characterizations of Biochars

The biochars were prepared using poplar leaves as raw materials and KHCO_3_ as the activator using an oxygen-limited pyrolysis method (Figure 1). The yield of biochar (PKC-0) without activation is 21.3 wt%, which is the highest among all of the samples. With the increase of the KHCO_3_ addition, the yields decline continuously (Appendix A). This implies that KHCO_3_ and its decomposition products react with the biomass and etch some biochar, resulting in a lower yield.

The morphology of biochars was observed by scanning electron microscopy (SEM), as shown in Figure 1; the biochar PKC-0 inherited the structure of the poplar leaf and presented a smooth and compact surface. After adding KHCO_3_, the biochars became small pieces and presented many pores, indicating KHCO_3_ has a significant pore-forming effect on biochar. With the increase of the KHCO_3_ addition, the biochars became increasingly fragmented, implying that the etching effect of KHCO_3_ on biochar is more and more obvious, which is consistent with the declining trend of biochar yields.

Raman spectra were performed to analyze the graphitization degree of these biochars. As shown in Figure 2, all patterns display two distinct peaks, which can be assigned to D-band (around 1340 cm^−1^) and G-band (around 1590 cm^−1^), corresponding to amorphous carbon and graphitic carbon in biochars, respectively [41]. The intensity ratio (*I*_G_/*I*_D_) of G-band to D-band was used to reflect the graphitization degree of biochars [42]. As shown in Appendix A, with the increase of the KHCO_3_ addition, the *I*_G_/*I*_D_ ratios decrease from 0.993 to 0.850, indicating that KHCO_3_ destroys the biochar matrix and reduces its graphitization degree.

X-ray photoelectron spectra (XPS) were performed to analyze the composition and the state of the elements of these biochars. The full spectra (Appendix A) show that biochars are mainly composed of carbon, nitrogen, and oxygen elements. As shown in Appendix A, with the increase of the KHCO_3_ addition, the carbon elemental atomic percentage keeps decreasing, while the oxygen elemental atomic percentage keeps rising, indicating that KHCO_3_ can oxidize carbon atoms and produce oxygen-containing groups. To understand the detailed bonding configuration of the oxygen element, high-resolution O1s spectra were recorded (Appendix A). The splitting O1s peaks of biochars at 530.4, 531.7, and 532.8 eV are attributed to the lattice oxygen (C=O) [43], carboxyl oxygen (−COOH), and hydroxy oxygen (−OH) [44,45], respectively. Among them, the proportion of carboxyl oxygen and hydroxyl oxygen is more than 90%, which may promote the adsorption performance of biochars.

In order to clarify the differences of pore structures between different biochars, the specific surface areas and pore size distributions were analyzed using nitrogen adsorption and desorption technology. As shown in Figure 3a, the nitrogen adsorption capacity of PKC-0 is very weak, indicating that its specific surface area is small, and its pore structure is not developed. Meanwhile, the nitrogen adsorption capacity of PKC-1 is much higher, indicating that KHCO_3_ plays a significant role in activating biochars. In the *P*/*P*_0_ range of less than 0.1, the nitrogen adsorption capacities of all biochars are relatively high, indicating that micropores are present in all biochars [46]. In addition, all biochars exhibit a type IV curve with hysteresis loops in the *P*/*P*_0_ range of 0.40–1.0, indicating the presence of mesopores [47].

The BET method and the BJH method were used to calculate the total specific surface area and the mesoporous specific surface area of biochars, respectively. The detailed calculation method is provided in the Appendix A, and the results are shown in Table 1. With the increase of the KHCO_3_ addition, both the total specific surface area and the mesoporous specific surface area increased initially, then stabilized, and finally decreased slightly, which implies that KHCO_3_ can effectively make pores in biochars, but excessive KHCO_3_ will cause the pore structure of biochars to collapse and reduce their specific surface areas. It should be noted that the total specific surface area of PKC-3 is close to that of PKC-2, but its mesoporous specific surface area is much higher than that of PKC-2. This change also applies to pore volume and average pore size. These results indicate that mesopores are probably formed by KHCO_3_ corrosion micropores. For the biochars activated by KHCO_3_, the ratios of mesoporous specific surface area to total specific surface area (*S*_meso_/*S*_total_) initially increased from 16% to approximately 65%, and then stabilized, which implies that excessive KHCO_3_ does not help the formation of the mesoporous structure. Figure 3b displays the pore size distributions of biochars calculated using the BJH method through nitrogen desorption isothermal curves. It shows that the pore sizes are all concentrated around 3.8 nm, further indicating the good mesopore-forming effect of KHCO_3_. All these characteristics demonstrate that KHCO_3_ can create both micropores and mesopores in biochars, thus significantly increasing their specific surface areas.

The interaction of KHCO_3_ with biochar can be divided into three steps. First, at 100–200 °C, KHCO_3_ decomposes to generate K_2_CO_3_ and CO_2_, which moderately erode the poplar leaf. Second, at 200–700 °C, the poplar leaf decomposes constantly, and K_2_CO_3_ is reduced to metallic potassium by carbon, which is etched into CO to create pores (K_2_CO_3_ + 2C → 2K + 3CO). Third, at 800 °C, K_2_CO_3_ decomposes slowly to generate K_2_O and CO_2_ (K_2_CO_3_ → K_2_O + CO_2_), which might etch carbon (CO_2_ + C → 2CO) to generate more pores and enlarge the existing pores.

### 2.2. Adsorption Kinetics

To evaluate the adsorption kinetics of biochars, the initial OTC concentration was fixed at 200 ppm, and the adsorption time was limited to 1–180 min. As shown in Figure 4, OTC is rapidly adsorbed by biochars at the initial stage of the adsorption process. As the adsorption proceeds, OTC continues to occupy the adsorption site, resulting in a decrease in the adsorption rate and finally reaching the adsorption equilibrium. It is worth noting that within 20 min, the adsorption capacities of PKC-3 and PKC-5 reached more than 80% of their maximum adsorption capacities, and PKC-4 reached more than 95% of its maximum adsorption capacity, which indicates that the adsorption rate of these biochars is very high. This is probably due to the presence of abundant mesopores in these biochars, which is beneficial to the diffusion of OTC and its interaction with adsorption sites in biochars [48,49,50]. Comparatively, within 20 min, the adsorption capacity of PKC-2 was only approximately 57% of its maximum adsorption capacity, which indicates that the adsorption rate of PKC-2 is much lower, even though its specific surface area is close to the above three biochars. This is probably due to its underdeveloped mesoporous structure, which hinders the diffusion of OTC. In addition, with the increase of the KHCO_3_ addition, the maximum adsorption capacities of biochars for OTC initially increased, and then decreased slightly. This change trend is the same as that of the total specific surface area of biochars, indicating that the total specific surface area may be an important factor for determining adsorption capacities.

To better understand the adsorption kinetics of these biochars, a pseudo-first-order model and a pseudo-second-order model were used to analyze the kinetic data—the fitted curves are displayed in Appendix A and the related parameters are displayed in Table 2. For all biochars, the correlation coefficient *R*^2^ of the pseudo-second-order model was higher than that of the pseudo-first-order model, indicating that the pseudo-second-order model provides a better fit to the kinetic data. These results imply that the adsorption interaction of OTC with biochars involves both physical adsorption and chemical adsorption [51]. Compared with all biochars, PKC-4 shows the highest adsorption performance; its maximum adsorption capacity of 543 mg/g was calculated from the pseudo-second-order model, which is higher than that reported in most of the literature (Appendix A), confirming that PKC-4 is highly effective for the adsorption of OTC.

Furthermore, an intra-particle diffusion model was used to study the role of diffusion in adsorption. As shown in Appendix A, these data can be mapped along three straight lines with different slopes and intercepts, indicating that the adsorption process can be divided into three stages. Stage I was the fastest step, which can be associated with surface diffusion. At stage I, there were abundant unoccupied adsorption sites on the surface of biochars, and the mass transfer resistance through the surface film was low, resulting in a high adsorption rate. At stage II, the surface adsorption sites were occupied, and adsorbate molecules faced higher mass transfer resistance to traverse the mesopores of biochars, resulting in a lower adsorption rate. At stage III, the adsorption rate was further reduced, reflecting the saturation of biochars without future adsorption. In addition, the intercepts of all lines were not equal to zero, indicating that the adsorption rate was controlled by film diffusion and intra-particle diffusion [52]. With the increase of the KHCO_3_ addition, the intercepts generally increase, indicating that the influence of the boundary layer on adsorption is more and more obvious.

### 2.3. Adsorption Isotherms

To evaluate the adsorption isotherms of biochars, the initial OTC concentration was fixed at 25 ppm, 50 ppm, 100 ppm, 150 ppm, and 200 ppm, respectively, and the adsorption time was fixed at 180 min. As shown in Figure 5, with the increase of the initial concentration of OTC, the equilibrium adsorption capacities of biochars and the equilibrium concentrations of OTC in the solution also increase.

To better understand the adsorption isotherms of biochars, the Langmuir model and the Freundlich model were used to analyze the equilibrium data—the fitted curves are displayed in Appendix A (Langmuir model) and Appendix A (Freundlich model), and the related parameters are displayed in Table 3. The correlation coefficient *R*^2^ of the Freundlich model was higher than that of the Langmuir model except for PKC-2 adsorption, indicating that the Freundlich model provides a better fit to the equilibrium data except for PKC-2 adsorption. The closer fit of the Freundlich model implies that the adsorption of OTC on biochars is mainly multi-layer adsorption [53]. In general, the factor 1/*n* is used to predict the adsorption difficulty. As shown in Table 3, the values of 1/*n* in all biochars are between 0 and 1, suggesting that the adsorption of biochars for OTC is favorable [54]. Furthermore, these biochars possess a mesoporous structure, and the proportion of mesoporous specific surface area of some biochars exceeds 64%, resulting in the accumulation of adsorbed molecules and ensuring the possibility of multi-layer adsorption. Comparatively, the Langmuir model provided a better fit to the equilibrium data for PKC-2 adsorption, indicating that the leading part of OTC adsorption is monolayer adsorption. As previously described, PKC-2 is rich in microporous structures, and the proportion of mesoporous specific surface area of PKC-2 is only 31%, which is not conducive to the accumulation of adsorbed molecules, making it difficult to achieve multi-layer adsorption.

### 2.4. Effects of Solution pH and Metal Ions

Solution pH is considered to be an important factor in the adsorption process, which affects the speciation of the pollutant and the surface charge of the adsorbent. To evaluate the impact of the solution pH, PKC-4 was selected as the adsorbent, and the initial OTC concentration was fixed at 200 ppm. As shown in Figure 6a, with the increase of the solution pH, the adsorption capacity decreased gradually, but the overall change was small, indicating that the effect of the solution pH was small. When the pH was lower, PKC-4 was electrically neutral, no obvious electrostatic repulsion between OTC and PKC-4 is observed. As the solution pH increases, the oxygen-containing functional groups (such as carboxyl group and hydroxyl group) on the surface of PKC-4 may be deprotonated, resulting in a negative surface charge of PKC-4. At the same time, OTC exists in the form of a negative charge, and the electrostatic repulsion between PKC-4 and OTC leads to the reduction of adsorption capacity. In addition, high pH value weakens the hydrogen bonding between PKC-4 and OTC, further reducing the adsorption capacity. It is noteworthy that the adsorption capacities are all around 500 mg/g in the range of 3–11 pH, indicating that PKC-4 can maintain high adsorption performance at different pH values.

The effect of metal cations in solution on adsorption equilibrium was also analyzed. For comparison, the concentration of metal cations was fixed at 100 ppm. As shown in Figure 6b, the presence of Na^+^ increased adsorption capacity, which may be because Na^+^ enhances the hydrophobic interaction between PKC-4 and OTC through salting out. K^+^ reduced adsorption capacity, possibly because it occupies the adsorption site of PKC-4. Compared with K^+^, Cu^2+^ had a more obvious effect on reducing adsorption capacity, which may be due to its high charge occupying more adsorption sites of PKC-4. Ca^2+^ and Mg^2+^ had little effect on adsorption capacity, which may be attributed to two aspects. On the one hand, they may occupy the adsorption sites of PKC-4 and reduce adsorption capacity; on the other hand, they may play a bridging role between OTC and the surface groups of PKC-4, increasing adsorption capacity [55]. Notably, Fe^3+^ reduced adsorption capacity by almost half. The reason may be that excessive positive charges occupy a large number of adsorption sites of PKC-4.

In addition, we analyzed the adsorption performance of PKC-4 in campus lake water and city water. As shown in Appendix A, compared to deionized water, the adsorption capacity of PKC-4 in campus lake water and city water decreased by 7.6% and 12.8%, respectively, indicating that PKC-4 has good potential in practical application.

### 2.5. Reusability

Recycling performance is an important factor in practical application. Here, biochar PKC-4 was regenerated by pyrolysis at 800 °C for 2 h under nitrogen flow, and then directly used for the next adsorption. As shown in Appendix A, after four cycles, adsorption capacity only decreased by approximately 6%, indicating that PKC-4 is highly stable in the adsorption process.

### 2.6. Possible Adsorption Mechanism

Various interactions participate in the adsorption of OTC by biochar. First, pore-filling is considered to be the main factor determining OTC adsorption. The correlation between the total specific surface area, pore volume, and the adsorption capacity of PKC-4 is shown in Appendix A. Both parameters display a good linear correlation with adsorption capacity, indicating that the pore structure has a very significant effect on adsorption. In addition, the nitrogen adsorption and desorption analysis shows that the total specific surface area and pore volume of PKC-4 are significantly reduced after use (Appendix A), demonstrating that pore-filling is an important factor determining OTC adsorption [52]. It is noted that the ratio *S*_meso_/*S*_total_ of PKC-4 after use increases from 64.4% to 96.8%, and the corresponding average pore size increases from 2.95 nm to 3.61 nm, indicating that both micropores and mesopores participate in the adsorption of OTC.

Second, the electrostatic interaction plays an important role in the adsorption of OTC by biochar. OTC exhibits different charges under different pH conditions, and the adsorption capacity of PKC-4 varies with the change of the solution pH. Furthermore, the capacity declines sharply in the presence of Cu^2+^ and Fe^3+^, demonstrating that the electrostatic interaction is one of the adsorption mechanisms.

Third, functional groups on biochar surfaces also promote the adsorption of OTC by biochar. XPS spectra confirm the existence of oxygen-containing functional groups in these biochars, which can generate H-bonds, and that OTC has hydroxyl and amino groups, which could also form H-bonds. Fourier transform infrared (FTIR) spectra show that the tensile vibration of O–H bond at 3627 cm^−1^ and the stretching vibration of C=O bond at 1640 cm^−1^ of PKC-4, shift to 3622 cm^−1^ and 1644 cm^−1^, respectively, after OTC adsorption (Appendix A). This is due to the deprotonation of carboxyl and hydroxyl groups in the adsorption process, indicating that the H-bonds play an important role in the adsorption process.

Finally, the adsorption of OTC by biochar can also be explained based on the π-π interaction. Raman spectra confirm the existence of graphitic carbon in these biochars, which can act as electron donors for their aromatic rings. These biochars can form a π-π electron-donor-acceptor (EDA) interaction with OTC for its electron-deficient polar aromatic rings.

## 3. Materials and Methods

### 3.1. Materials and Reagents

Poplar leaves were collected in the campus of Shandong First Medical University and Shandong Academy of Medical Sciences. Potassium bicarbonate, hydrochloric acid, sulphuric acid, sodium hydroxide, sodium chloride, potassium carbonate, calcium chloride, magnesium sulfate, copper nitrate, and iron nitrate were purchased from Sinopharm Chemical Reagent Co., Ltd. (Beijing, China). Nitrogen was purchased from Taian Yingchun Gas Co., Ltd. (Taian, China). Oxytetracycline hydrochloride was purchased from Beijing InnoChem Science and Technology Co., Ltd. (Beijing, China). All purchased reagents were used without further purification.

### 3.2. Preparation of Biochar

Poplar leaves were cleaned, dried, smashed to pieces, and stored in a dry atmosphere. A total of 5 g of smashed poplar leaf was mixed carefully with KHCO_3_ (the weight 1~5 g), and then the mixtures were placed in a tubular furnace and pyrolyzed at 800 °C for 2 h under nitrogen flow at a heating rate of 10 °C per minute. The samples were collected after cooling to room temperature, and washed with 2 M hydrochloric acid for 4 h at room temperature to remove inorganic salts. After filtration, washing, and drying, the biochar was obtained, which was denoted as PKC-*n*, where *n* represents the weight of KHCO_3_ added in the synthesis process.

### 3.3. General Procedure for the Adsorption of Oxytetracycline Hydrochloride

A total of 10 mg of biochar was added into 30 mL of OTC solution with a certain concentration, then placed into a table concentrator operating at a speed of 150 r/min at 25 °C. After adsorption for a period of time, some of the aqueous mixture was collected, and then diluted to 3 mL and filtered with 0.22 μm membrane. Its absorbance at 275 nm was analyzed using a UV-visible spectrophotometer. Each dataset was repeated three times. The adsorption capacity of biochar to OTC was calculated using the following equation:(1)$$q=\frac{\left(A_{0}-A_{t}\right)\times c_{0}\times V}{m}$$
where *q* (mg/g)—adsorption amount of OTC at time *t*; *A*_0_—absorbance of initial aqueous mixture; *A_t_*—absorbance of aqueous mixture at time *t*; *c*_0_ (mg/L)—initial OTC concentration; *V* (*L*)—volume of OTC solution; and *m* (g)—mass of biochar.

### 3.4. Analysis of Adsorption Kinetics and Isotherms

The adsorption kinetics were fitted with a pseudo-first-order model (Equation (2)), a pseudo-second-order model (Equation (3)) and an intra-particle diffusion model (Equation (4)), respectively:(2)$$q=q_{e}\left(1-e^{-k_{1}t}\right)$$
(3)$$q=\frac{q_{ek_{2}t}^{2}}{1+q_{e}k_{2}t}$$
(4)$$q=k_{p}t^{1/2}+b$$
where q (mg/g)—adsorption amount of OTC at time *t*; *q*_e_ (mg/g)—equilibrium adsorption amount of OTC; *k*_1_—adsorption constant of pseudo-first-order model; *k*_2_—adsorption constant of pseudo-second-order model; and *k_p_*—adsorption constant of intra-particle diffusion model.

The adsorption isotherms were fitted with the Langmuir model (Equation (5)) and the Freundlich model (Equation (6)):(5)$$\frac{c_{e}}{q_{e}}=\frac{1}{q_{m}K_{L}}+\frac{c_{e}}{q_{m}}$$
(6)$$\ln q_{e}=\ln K_{F}+\frac{1}{n}\ln c_{e}$$
where *q*_e_ (mg/g)—equilibrium adsorption amount of OTC; *c*_e_ (mg/g)—equilibrium OTC concentration; *q*_m_ (mg/g)—the maximum adsorption amount of OTC; *K*_L_—the Langmuir constant; *K*_F_—the Freundlich affinity coefficient; and *n*—the Freundlich exponential coefficient.

### 3.5. Characterizations

The morphology of biochar was observed by JSM-7610F SEM (Jeol, Tokyo, Japan). Nitrogen adsorption and desorption isotherms were analyzed at 77 K using a Jingweigaobo JW-BK100C analyzer (Beijing, China), and the samples were degassed at 200 °C for 4 h before measurements. The absorbance of the aqueous mixture was analyzed using a UV-1800 UV-Vis spectrophotometer (Shimadzu, Kyoto, Japan). FTIR spectra were performed on an IRPRestige-21 infrared spectrometer. Raman spectra were recorded using a renishaw inVia Raman spectrometer with an Ar laser (532 nm) as the excitation source. XPS were obtained from a ULVAC-PHI PHI5000 VersaprobeIII instrument (ULVAC-PHI, Chigasaki, Japan) with Al Kα radiation.

## 4. Conclusions

In this study, we prepared low-cost biochars through pyrolysis with poplar leaves as the raw material and KHCO_3_ as the activator. Characterizations indicate that KHCO_3_ can create abundant mesopores in biochars by etching micropores, but excessive KHCO_3_ cannot continuously promote the formation of a mesoporous structure. When used for adsorption of OTC, PKC-4 exhibits the best adsorption performance due to its large surface area and rich mesoporous structure. Model analysis shows that the pseudo-second-order model and the Freundlich model can best describe the adsorption kinetics and isotherms. In addition, PKC-4 displays high OTC adsorption capacity in a wide pH range, different metal ions, and different water matrices. Moreover, the biochar can be regenerated easily, and its adsorption capacity decreases little after four cycles. The adsorption mechanism is mainly pore-filling, though electrostatic interaction, hydrogen bond, and π-π interaction are also involved. This study realizes biomass waste recycling and highlights the potential of poplar leaf-based biochar for the adsorption of antibiotics.

## Data Availability

Data is contained within the article or Appendix A.

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
