# Peer review of "Preparation of Biochar with Developed Mesoporous Structure from Poplar Leaf Activated by KHCO3 and Its Efficient Adsorption of Oxytetracycline Hydrochloride"

_molecules, 2023, doi:10.3390/molecules28073188_

Round 1

Reviewer 2 Report

In this manuscript, low-cost biochars through pyrolysis with poplar leaves as materials and KHCO3 as activator have been designed and prepared. This work is helpful to realize the resource utilization of biomass waste and improve the potential of poplar leaf-based biochar to adsorb antibiotics. However, there are still some problems that need further modification.

1.         It is better to provide one scheme to present the whole experimental process for better understanding of this work.

2.         The SEM pictures in Figure 1 is not clear. Please supplement new SEM pictures clearly.

3.         All data were analyzed in point-and-line format to facilitate data comparison and analysis.

4.         A comparison table needs to be added to compare the adsorption properties with those of other biomass materials.

5.         When introducing the water pollution or water treatment, some recent and important articles should be included: Synthesis and Application of Granular Activated Carbon from Biomass Waste Materials for Water Treatment: A Review; Biochar derived from non-customized matamba fruit shell as an adsorbent for wastewater treatment; etc.

6.         Are all the experimental data single? Three parallel experiments are needed for error analysis and error bars are added.

7.         The analysis content of material adsorption mechanism is lacking, which needs to be included in the manuscript.

8.         The different water pollution and treatments should be introduced with supporting articles, such as fluoride (New Journal of Chemistry 46, 490-497, 2022; Polymers 14 (24), 5417, 2022); uranium (e-Polymers 22 (1), 399-410, 2022); Ph2+ (Vacuum 189, 110229, 2021); dye (Chemical Engineering Journal 446, 136851, 2022; ACS nano 15 (12), 20666–20677, 2021); etc.

9.         There are still some typos and grammar issues in the manuscript. Authors should carefully recheck the whole manuscript.

10.     There are too many too old references, which is better to be deleted or replaced with recent articles to show the novelty of this work.

Round 2

Reviewer 1 Report

Dear authors,

I accept your corrections in manuscript but I have one suggestions to Figure 1. Could you write not (zoom 5000) but (x5000)? Subsequently, the manuscript could be published in revised form. 
